# Endophytic Fungal Diversity of Mangrove Ferns *Acrostichum speciosum* and *A. aureum* in China

**DOI:** 10.3390/plants13050685

**Published:** 2024-02-29

**Authors:** Hongjuan Zhu, Wending Zeng, Manman Chen, Dan He, Xialan Cheng, Jing Yu, Ya Liu, Yougen Wu, Dongmei Yang

**Affiliations:** 1School of Breeding and Multiplication (Sanya Institute of Breeding and Multiplication), Hainan University, Sanya 572025, China; zhuhongjuan1013@163.com (H.Z.); damianceng@163.com (W.Z.); cm2713547954@163.com (M.C.); hdd6392@163.com (D.H.); yujinghxy@163.com (J.Y.); liuya113@hainanu.edu.cn (Y.L.); wygeng2003@163.com (Y.W.); 2School of Tropical Agriculture and Forestry, Hainan University, Danzhou 571737, China; 3School of Life Science and Technology, Lingnan Normal University, Zhanjiang 524048, China; chengxl@lingnan.edu.cn

**Keywords:** *Acrostichum speciosum*, *A. aureum*, mangrove, endophytic fungi, ITS, conservation

## Abstract

Microbial communities are an important component of mangrove ecosystems. In order to reveal the diversity of endophytic fungi in the mangrove ferns *Acrostichum speciosum* and *A. aureum* in China, the internal transcribed spacer (ITS) regions of endophytic fungi in four plant tissues (leaves, petioles, roots, and rhizomes) from three locations (Zhanjiang, Haikou, and Wenchang) were sequenced. The richness, species composition, and community similarity were analyzed. The main results are as follows: the dominant fungi in *A. speciosum* and *A. aureum* belonged to the phyla Ascomycota and Basidiomycota, accounting for more than 75% of the total identified fungi; in terms of species composition at the operational taxonomic unit (OTU) level, the endophytic fungi in *A. aureum* were more diverse than those in *A. speciosum*, and the endophytic fungi in rhizomes were more diverse than in other tissues. In Zhanjiang, both *A. speciosum* and *A. aureum* showed the richest diversity of endophytic fungi, both at the OTU classification level and in terms of species composition. Conversely, the richness of endophytic fungi in the samples of *A. speciosum* from Wenchang and Haikou is extremely low. The regional differences in dominant fungi increase with the degrading of taxonomic levels, and there were also significant differences in the number of unique fungi among different origins, with Zhanjiang samples having a larger number of unique fungi than the other locations. There were significant differences in the dominant fungi among different tissues, with Xylariales being the dominant fungi in rhizomes of *A. speciosum* and Hypocreales being the dominant fungi in the petioles, roots, and rhizomes of *A. aureum*. Overall, the community similarity of endophytic fungi among locations is moderately dissimilar (26–50%), while the similarity between tissues is moderately similar (51–75%). The low diversity of endophytic fungi could be one of the main reasons for the endangerment of *A. speciosum*. The protection of the diversity of endophytic fungi in the underground parts of *A. speciosum* is essential for the conservation of this critically endangered mangrove fern.

## 1. Introduction

Mangroves are distinct and diverse ecosystems predominantly located along tropical and subtropical coastlines, specifically in intertidal zones. They possess abundant biodiversity and can sustain the existence of aquatic and terrestrial organisms [1,2]. Mangroves play a vital role in enhancing intertidal environments, offering protection against wind and waves, and augmenting the biodiversity of coastal areas. These ecosystems are commonly referred to as “forests of the sea” [3,4,5,6] and hold substantial significance in sustaining global marine and biological resources [7,8].

According to a survey conducted by the Food and Agriculture Organization of the United Nations (FAO), mangroves are currently facing a critical situation. It is estimated that over three quarters of the world’s mangroves are under threat, posing a significant risk to the delicate balance they sustain. The global mangrove ecosystem remains highly vulnerable, with mangroves distributed along the coastlines of 123 countries. Over the past 40 years, it is estimated that more than 20% of mangroves worldwide have disappeared due to human activities and natural decline. Despite the loss of 677,000 hectares of mangroves between 2000 and 2020, the rate of mangrove loss has slowed down by nearly a quarter (23%) in the second decade of this century (https://news.un.org/zh/story/2023/07/1120107 (accessed on 25 September 2023)). In China, mangroves are primarily found in Guangdong, Guangxi, Hainan, Fujian, and Taiwan provinces. Over the past 40 years, the total area of mangroves in China has shown a fluctuating trend. From 1978 to 2000, the total area of mangroves decreased from approximately 30,000 hectares to 20,000 hectares. However, from 2000 to 2018, there was an increase of 5700 hectares in the total mangrove area. With strengthened efforts in mangrove conservation, the rate of mangrove area reduction has gradually slowed down in recent years [9]. Nevertheless, these rare and fragile mangrove ecosystems show high productivity in China and all over the world [10,11,12].

In addition to the presence of plant and animal communities, microbial communities also constitute a significant component of mangrove ecosystems [13]. Given the recurrence of tidal flooding, these ecosystems undergo frequent fluctuations in salinity and nutrient levels. As such, microbial communities play a pivotal role in the biogeochemical cycling of nutrients and the regulation of associated properties within mangrove ecosystems [14]. The primary mechanism involved in nutrient transformation in these ecosystems is microbial activity [14,15]. The intertwined interactions among microorganisms collectively uphold the chemical, nutritional, and ecological equilibrium of mangrove ecosystems. Moreover, the composition and structure of microbial communities can be influenced by the ecological environment and geographical location [16]. Consequently, microbial communities can serve as valuable indicators for monitoring the ecological environment of mangroves [16].

In recent times, mangroves have emerged as a pivotal area of focus for investigating fungal biodiversity across both marine and terrestrial environments [13]. Among the various microorganisms present, endophytic fungi play a significant role, as they reside within the tissues of plants for a specific duration, or throughout their entire lifecycle, without imposing any harm upon their host plants [17]. Despite certain endophytic fungi being cultivable, the inconsistent growth rates and limitations associated with conventional isolation techniques pose challenges in isolating the majority of endophytic fungi. As a result, molecular biology techniques have been widely adopted to detect the diversity of endophytic fungi within plant tissues [18]. Recently, the internal transcribed spacer (ITS) region has gained broad usage in assessing the community diversity of endophytic fungi [18,19,20,21]. The high-throughput sequencing capabilities of ITS facilitate the acquisition of abundant sequence information pertaining to endophytic fungi, enabling comprehensive analysis [22]. Consequently, ITS technology can effectively identify all endophytic fungi present within plant tissues, particularly those that exhibit slow growth rates or exhibit difficulties in cultivation on artificial culture media.

The species *Acrostichum speciosum* and *A. aureum*, both belonging to the family Pteridaceae, are ferns living in mangroves and are important components of mangrove ecosystems [23] (https://www.iplant.cn/info/Acrostichum%20speciosum (accessed on 28 September 2023)). Compared to the woody plants in mangroves, they have a shorter growth cycle and mainly live in the intertidal zone of the coast, making them sensitive to environmental changes [24,25,26]. Therefore, they can serve as important ecological indicators for monitoring the health of mangroves. There are differences in morphology, distribution range, habitat, and population size between *A. speciosum* and *A. aureum* in China. *A. aureum* is widely distributed in the pan-tropical region, with larger leaves and a preference for sunlight. It usually grows at the edge of mangroves as well as on abandoned farmland and marshes that are near mangroves. However, *Acrostichum speciosum* has narrow leaves and a limited distribution range, only in tropical and subtropical regions. It prefers shade and only grows in shaded environments under the mangrove canopy [27,28]. *Acrostichum speciosum* is naturally distributed in only two locations in China, which are Wenchang, Hainan, and Zhanjiang, Guangdong. After the establishment of the Dongzhaigang Mangrove Nature Reserve in Haikou, *Acrostichum speciosum* was introduced from Wenchang. Its populations are small, and thus it is classified as critically endangered (CR) in China. There is little research on *A. speciosum*, mainly focusing on its morphology and the secondary metabolites of its endophytic fungi [29,30,31]. In contrast, there is relatively more research on *A. aureum*, such as with regard to its chloroplast genomes, the antioxidant activities of secondary metabolites from endophytic fungi, and its role in mangrove conservation [32,33,34,35,36,37]. However, the diversity of endophytic fungi in *A. speciosum* and *A. aureum* is still unclear. Whether the diversity of endophytic fungi is critical to the endangered status of *A. speciosum* remains unknown.

This study examines the variation in endophytic fungi between the critically endangered plant *A. speciosum* and its closely related species *A. aureum*, adopting a microbiological standpoint and primarily employing ITS technology. The objective is to offer novel perspectives on the conservation of *A. speciosum* and the safeguarding of mangrove ecosystems, with potential avenues for investigating the correlation between the abundance of these fungi and the distinctive characteristics of mangroves.

## 2. Results

### 2.1. Operational Taxonomic Units Clustering and Species Annotation

A grand total of 6,155,982 raw reads were acquired, out of which 6,102,015 clean reads were obtained post filtration. This comprised 3,085,784 clean reads for *A. speciosum* and 3,016,231 for *A. aureum*. Subsequently, a total of 5,565,124 high-quality tags were obtained after merging the reads and implementing rigorous quality control measures to eliminate chimeric sequences. Regarding the sequencing data, the Q20 value surpassed 99.7%, the Q30 value surpassed 98.8%, and the GC content ranged from 37.7% to 59.6% (Appendix A).

The rarefaction curves, generated from samples obtained from distinct tissues of *A. speciosum* and *A. aureum*, exhibited an initial steep ascent followed by a gradual plateauing, indicative of reaching a saturation point with increasing sequencing depth. This observation suggests that the sequencing data adequately encompass the entirety of the endophytic fungi present within the samples (Figure 1).

### 2.2. Species Composition of Endophytic Fungi

A total of 3065 operational taxonomic units (OTUs) of endophytic fungi were obtained in this sequencing, with 1944 OTUs from *A. speciosum* and 2498 OTUs from *A. aureum*. There were 1377 OTUs shared by *A. speciosum* and *A. aureum* (Figure 2A). The endophytic fungi belonged to 394 genera, 212 families, 90 orders, 38 classes, and 13 phyla. The endophytic fungi from *A. speciosum* belonged to 296 genera, 169 families, 78 orders, 30 classes, and 11 phyla, while those in *A. aureum* were from 353 genera, 201 families, 85 orders, 37 classes, and 13 phyla (Figure 2B,C). In terms of species composition at the OTU level, the endophytic fungi of *A. aureum* were more abundant than those of *A. speciosum*.

At the phylum level, apart from the unidentified species, more than 80% of endophytic fungi in *A. speciosum* and *A. aureum* were found to belong to Ascomycota and Basidiomycota, indicating that they are the dominant phyla in *A. speciosum* and *A. aureum* (Figure 3A,B). At the class level, the dominant fungal strains in *A. speciosum* and *A. aureum* were Sordariomycetes, Dothideomycetes, and Eurotiomycetes (Figure 3C,D). At the order level, the dominant fungal strains in *A. speciosum* and *A. aureum* were Capnodiales and Hypocreales, but the proportion of Capnodiales was higher (Figure 3E,F). At the family level, more than 50% belonged to unknown families, and Mycosphaerellaceae, Nectriaceae, and Herpotrichiellaceae were the dominant families among the known endophytic fungi, though there were significant differences in their proportions between *A. speciosum* and *A. aureum* (Figure 3G,H). At the genus level, more than 70% of the fungi remained unidentified. Among the known endophytic fungi, the dominant fungal genera between *A. speciosum* and *A. aureum* were different. The dominant fungal genera in *A. speciosum* were *Fusarium* (3%), *Cladophialophora* (2%), and *Dactylonectria* (2%), while *Dactylonectria* (10.17%), *Fusarium* (2.25%), and *Pseudocercospora* (2.15%) were dominant in *A. aureum* (Figure 3I,J).

In conclusion, at higher taxonomic levels such as phylum and class, the overall composition of dominant endophytic fungi in *A. speciosum* and *A. aureum* was not significantly different. However, at the family and genus levels, there were significant differences in the composition of dominant endophytic fungi, and the proportions of unidentified endophytic fungi were high.

### 2.3. Factors Influencing Endophytic Fungal Richness and Species Composition

#### 2.3.1. Influence of Locations

At the out level, the highest richness of *A. speciosum* was found in Zhanjiang, reaching 1388 OTUs, while the lowest was in Wenchang, with only 871 OTUs (Figure 4A). As for *A. aureum*, samples from Zhanjiang also had the largest number, up to 1910 OTUs (Figure 4B). In terms of identified endophytic fungi, *A. speciosum* from Zhanjiang showed the most complex composition, with 236 genera, 139 families, and 64 orders, while the samples from Wenchang showed the simplest species composition. Similarly, *A. aureum* in Zhanjiang exhibited the highest diversity, with 308 genera, 177 families, and 77 orders. It can be seen that both *A. speciosum* and *A. aureum* had the most complex species composition of endophytic fungi in Zhanjiang, and at the same location, the endophytic fungi of *A. speciosum* were less diverse than those of *A. aureum*.

The results obtained from ITS sequencing analysis (Appendix A) demonstrate notable geographical variations in the endophytic fungi composition between *A. speciosum* and *A*. *aureum.* In the case of *A. speciosum*, the dominant fungi at the phylum level in all three locations were Ascomycota, Basidiomycota, and Glomeromycota. However, at lower taxonomic levels such as class, order, family, and genus, substantial differences in dominant fungi were observed among the three locations, with the differences becoming more pronounced as the taxonomic level decreased. For instance, at the class level, the dominant fungi in samples from Zhanjiang were Sordariomycetes, Dothideomycetes, Tremellomycetes, Eurotiomycetes, and Glomeromycetes, while in Wenchang, they were Dothideomycetes, Sordariomycetes, and Agaricomycetes, and in Haikou, they were Dothideomycetes, Sordariomycetes, Glomeromycetes, and Agaricomycetes. At the family level, the dominant fungi in Zhanjiang were Nectriaceae, Herpotrichiellaceae, Bulleribasidiaceae, and Mycosphaerellaceae; in Wenchang, they were Mycosphaerellaceae; and in Haikou, they were Mycosphaerellaceae, Glomeraceae, and Herpotrichiellaceae. At the genus level, the dominant fungi in Zhanjiang were *Fusarium*, *Dactylonectria*, *Cladophialophora*, and *Neopestalotiopsis*; in Haikou, they were *Cladophialophora*; and in Wenchang, a higher abundance of unidentified fungi was found, with *Zasmidium*, *Pseudocercospora*, and *Clavulinopsis* being relatively dominant genera. It is worth noting that *Clavulinopsis* was exclusively present in Wenchang and did not rank among the top 15 most abundant genera. In the case of *A. aureum*, the influence of location on the dominant fungi was not evident at the phylum and class levels. However, as the taxonomic level decreased, the differences in dominant fungi among the three locations became more apparent. Specifically, at the phylum and class levels, samples from the three locations shared the same dominant fungi, which were identical to those observed in *A. speciosum*. At the family level, although several dominant fungi such as Nectriaceae and Mycosphaerellaceae were shared among the three locations, each location had its own dominant fungi, such as Herpotrichiellaceae in Zhanjiang, Chaetosphaeriaceae in Wenchang, and Diaporthaceae in Haikou. At the genus level, the differences in dominant fungi among the locations were particularly pronounced. In addition, there were obvious differences in the unique fungi among the three locations (Appendix A). Among the known endophytic fungi, 42 genera were found only in *A. speciosum* (18 genera in Zhanjiang, three genera in Haikou, and 10 genera in Wenchang), and 99 genera were found only in *A. aureum* (47 genera in Zhanjiang, six genera in Haikou, and three genera in Wenchang). Among the shared endophytic fungi of *A. speciosum* and *A. aureum*, 18 genera were found only in Zhanjiang, and only two species were found in Haikou and Wenchang. Therefore, the number of unique fungi in Zhanjiang is significantly higher than in other locations.

According to the richness of endophytic fungi in the top 30 subjects (Figure 5), it is clear that the species and quantity of endophytic fungi of *A. speciosum* and *A. aureum* in Zhanjiang are more abundant compared to the samples from the other two locations. The richness of different endophytic fungi varied with species and locations. For *A. speciosum*, the richness of endophytic fungi from Wenchang and Haikou was extremely low.

To further illustrate the impact of location on the composition of endophytic fungi, a ternary phase diagram was applied to compare the species richness of the top 15 families of endophytic fungi from the three locations. As shown in Figure 6, for *A. speciosum*, the relative abundance of the Zhanjiang samples was more concentrated compared to the Wenchang and Haikou samples; for *A. aureum*, the samples from the three locations exhibited a more evenly distributed pattern. This indicates that the endophytic fungi of *A. speciosum* have more local characteristics, and their species composition is more susceptible to the influence of location.

#### 2.3.2. Influence of Tissue Positions

At the OTU level, the OTU richness levels in different tissues of *A. speciosum* were rhizome (1191) > leaf (1030) > petiole (954) > root (907) (Figure 7A); for *A. aureum*, they were rhizome (1644) > leaf (1304) > root (1147) > petiole (991) (Figure 7B). In terms of species composition, the endophytic fungal species composition of *A. speciosum* was rhizome (203 genera, 127 families, 58 orders) > petiole (189 genera, 125 families, 57 orders) > leaf (187 genera, 126 families, 60 orders) > root (169 genera, 110 families, 53 orders); for *A. aureum*, it was rhizome (262 genera, 158 families, 69 orders) > leaf (230 genera, 151 families, 75 orders) > root (222 genera, 143 families, 68 orders) > petiole (181 genera, 123 families, 59 orders). It can be inferred that the endophytic fungal species composition at the OTU level is most abundant in the rhizome for both *A. speciosum* and *A. aureum*, and in the same tissue, the endophytic fungi of *A. aureum* are more complex than those of *A. speciosum*.

Because more than 50% of the unidentified endophytic fungi at the levels of order and genus were found in *A. speciosum* and *A. aureum*, the species composition of endophytic fungi in different tissues was analyzed at the family level. For *A. speciosum* (Figure 8A), Xylariales was the most abundant in rhizomes, and it was also distributed widely in leaves and petioles. Capnodiales was predominant in leaves and petioles, while Hypocreales was dominant in roots. For *A. aureum* (Figure 8B), the dominant endophytic fungi in leaves were the same as those in *A. speciosum*, and Hypocreales was predominant in petioles, roots, and rhizomes. Additionally, the rhizomes of both *A. speciosum* and *A. aureum* had the highest numbers of unidentified endophytic fungi. Above all, the dominant fungi in *A. speciosum* and *A. aureum* were similar. Based on the top 15 families of endophytic fungi at the family level (Figure 8), *A. speciosum* and *A. aureum* shared 11 common families, excluding the unidentified endophytic fungi.

The endophytic fungi in different tissues of *A. speciosum* and *A. aureum* overlapped, but the diversity of dominant fungi increased with the degrading of taxonomic levels. For example, at the order level (Appendix A), both *A. speciosum* and *A. aureum* had common dominant fungi in rhizomes, leaves, petioles, and roots. In rhizomes, the dominant fungi were Chaetothyriales and Hypocreales; in leaves, they were Capnodiales; in petioles, they were Xylariales; and in roots, they were Hypocreales. At the genus level, *A. speciosum* and *A. aureum* only had three common dominant fungi, *Cladophialophora*, *Dactylonectria*, and *Cladophialophora*, in rhizomes and roots, indicating a relatively low similarity.

Besides this, there were significant differences in the unique fungi in different tissues (Appendix A). Among the known endophytic fungi at the genus level, 13 genera of endemic fungi from *A. speciosum* were only distributed in rhizomes, while 10 genera, six genera, and four genera were distributed exclusively in leaves, leaf stalks, and roots, respectively. Among the 99 genera of endemic fungi in *A. aureum*, 25 genera were exclusively located in the rhizome, while 13 genera, seven genera, and five genera were only found in the roots, leaves, and leaf stalks, respectively. It can be concluded that rhizomes have more unique fungi than other tissues.

### 2.4. The Diversity Index and Community Similarity of Endophytic Fungi

#### 2.4.1. Alpha Diversity

Community diversity and species richness are commonly represented by diversity indices such as richness, ACE, chao1, Shannon’s, Simpson’s, and Pielou indices. Richness is a diversity parameter based on the resampling of each sample, which can help elucidate the degree of species diversity within a group intuitively, including maximum value, minimum value, and median. Based on the richness values at the OTU level (Figure 9), both *A. speciosum* and *A. aureum* had larger inter-group differences in the Zhanjiang samples compared to the Wenchang and Haikou samples. This indicates that the endophytic fungal species diversity in the Zhanjiang samples is higher. In addition, the richness values of the endophytic fungi in *A. aureum* were higher than those of *A. speciosum*. This suggests that the endophytic fungi in *A. aureum* are more abundant than in *A. speciosum*. According to the richness values of different tissues (Figure 10B), the inter-group differences in the rhizomes of *A. speciosum* and *A. aureum* were the highest, and the species richness of endophytic fungi in the rhizomes of *A. aureum* was significantly higher than in other tissues.

The alpha diversity indices, chao1 and ACE, can reflect the richness of a community, while Simpson’s and Shannon’s can reflect the diversity of a community, and Pielou reflects the evenness of a community. Larger values of chao1, ACE, Shannon, and Pielou indices imply a higher richness, diversity, or evenness in the species, while a larger value of Simpson indicates lower species abundance. Given the same species richness, a community with greater evenness is considered to have greater diversity. In our study, the species diversity, community richness, and species evenness of *A. aureum* were higher than those of *A. speciosum* (Table 1). Given the same species, the species diversity, richness, and evenness of the Zhanjiang samples were higher than those of other regions.

#### 2.4.2. Beta Diversity

At the OTU level, community dissimilarities among endophytic fungi from different locations were compared using the Bray–Curtis algorithm through the Principal Coordinate Analysis (PCoA) method. In the PCoA, smaller distances between samples indicate higher similarity in endophytic fungal populations, resulting in the clustering of similar samples. As depicted in Figure 10A, certain endophytic fungi from both *A. speciosum* and *A. aureum* exhibited clustering or overlapping, signifying a degree of similarity in their endophytic fungal composition. Specifically for *A. speciosum* (Figure 10B), the majority of samples from Zhanjiang exhibited a cohesive clustering, with only a minor overlap with samples from Haikou, while samples from Haikou and Wenchang demonstrated a relatively dispersed distribution. This pattern suggests lower similarity between the Zhanjiang samples and those from Haikou and Wenchang, aligning with the aforementioned results. In the case of *A. aureum* (Figure 10C), sample distribution appeared to be relatively uniform, with partial clustering observed among samples from Zhanjiang, Haikou, and Wenchang, indicating a certain level of similarity among the endophytic fungal communities from these three locations. Notably, the species composition of endophytic fungi in *A. speciosum* displayed greater geographic distinctiveness and susceptibility to location-specific influences compared to *A. aureum*. Furthermore, the endophytic fungi in different tissues of *A. speciosum* exhibited higher tissue specificity compared to those in *A. aureum*. For instance, nearly all endophytic fungi derived from the rhizomes of *A. speciosum* clustered together, with only a minimal overlap with those derived from the leaves (Figure 10D), while endophytic fungi from different tissues of *A. aureum* displayed more even distribution (Figure 10E).

#### 2.4.3. Community Similarity

According to Table 2 and Table 3, the similarity in the endophytic fungal community structure in *A. speciosum* and *A. aureum* varied across different locations and tissues. Overall, the similarity of endophytic fungi between locations ranged from 26% to 50%, indicating a moderate dissimilarity, while the similarity between tissues ranged from 51% to 75%, indicating a moderate similarity.

In terms of locations, given the same plant species, the lowest similarity of endophytic fungi in *A. speciosum* was found in samples from Zhanjiang and Wenchang, while the highest similarity was in the Zhanjiang and the Haikou samples; the lowest similarity of endophytic fungi in *A. aureum* was found between Zhanjiang and Haikou, and the highest similarity was found between Zhanjiang and Wenchang. For the same locations, the lowest similarity was in the Wenchang samples of *A. speciosum* and *A. aureum*, while the highest similarity was between the Zhanjiang samples of *A. speciosum* and *A. aureum*. Overall, the lowest similarity was between the Wenchang samples of *A. speciosum* and the Zhanjiang samples of *A. aureum*, while the highest similarity was between the Zhanjiang samples of *A. speciosum* and the Haikou samples of *A. aureum*.

In terms of different tissues, the lowest similarity of endophytic fungi in *A. speciosum* was found between petioles and rhizomes, while the highest similarity was between petioles and roots. For *A. aureum*, the lowest similarity of endophytic fungi was between petioles and rhizomes, while the highest similarity was between leaves and petioles. Overall, the lowest similarity was between the samples in the leaves of *A. speciosum* and the rhizomes of *A. aureum*, while the highest similarity was between the samples in the roots of *A. speciosum* and the petioles of *A. aureum*.

## 3. Discussion

Mangroves are a distinct class of host plants that possess a rich reservoir of endophytic fungal resources, constituting the second largest fungal community in the marine ecosystem [38]. The dominant genera of endophytic fungi found in mangroves are *Aspergillus*, *Penicillium*, and *Fusarium* [39]. While most endophytic fungi exhibit a broad host range, a small proportion of species display high host specificity. Moreover, the community structures of endophytic fungi in plants are subject to dynamic changes and are strongly influenced by geographical locations, environmental conditions, and different tissue compartments. Within a specific region, the composition of endophytic fungal species in the same plant species tends to be highly similar, although the degree of infection by endophytic fungi in various tissues and the species richness of endophytic fungi are influenced by the ecological environment and specific tissue compartments [40,41,42].

### 3.1. Relationships between Hosts and Endophytic Fungi

The most common endophytic fungi infecting higher plants are mainly from the phyla Ascomycota and Basidiomycota [43,44]. The identified endophytic fungi in our study also mainly belong to these two phyla. For *A. speciosum*, the phyla Ascomycota and Basidiomycota together account for 81% of the total identified endophytic fungi, while for *A. aureum*, they account for 75%. This indicates that the phyla Ascomycota and Basidiomycota are also dominant fungi in mangrove ferns.

Different plant species have a significant impact on the species composition of endophytic fungi [45]. Endophytic fungi need to penetrate the surface of the host and extract nutrients from the host’s interior. The nutritional composition of different host plants and plant tissues can affect the abundance and species composition of endophytic fungi [46,47]. In our study, in the same location and the same tissue, the endophytic fungi in *A. aureum* were more abundant than those in *A. speciosum*. This indicates that the host plant *A. aureum* and its internal nutritional composition are more suitable for the survival of endophytic fungi than *A. speciosum*.

Compared with *A. aureum*, the endophytic fungal diversity of *A. speciosum* is relatively low. This is likely to be one of the main reasons for the endangered status of *A. speciosum*. Endophytic fungi can form mutualistic symbiotic relationships with host plants, effectively promoting the growth of host plants and enhancing their resistance to biotic and abiotic stresses [48]. For example, endophytic fungi and halophytes can form symbiotic relationships in extreme high-salt environments, exhibiting high fungal community diversity and enhancing plant resistance to salt stress [49,50]. Therefore, the low diversity of endophytic fungi could result in a weak capacity of *A. speciosum* to adapt to environmental changes, thus leading to its endangered status.

### 3.2. Effects of Ecological Environments on the Abundance and Species Composition of Endophytic Fungi

The abundance and species composition of endophytic fungi often vary with changes in ecological environments. The diversity of endophytic fungal communities in urban and rural areas were investigated, and it was found that the endophytic fungi in urban forests were lower in abundance compared to those in rural forests. Furthermore, the species diversity of endophytic fungi in urban forests was negatively correlated with the degree of urbanization [51], which is consistent with the results of our study. Samples collected from Zhanjiang on the remote island of Jilongshan had the highest abundance of endophytic fungi and the most complex species composition, which is closely related to the degree of human disturbance. Due to the inconvenience of transportation, there is scarce human activity in Jilongshan, and the local vegetation is well protected. On the other hand, the samples from Haikou and Wenchang were located in natural reserves, which are also ecological tourist areas with well-developed infrastructure and convenient transportation. Thus, human activities there are frequent, resulting in significant disturbance to the local vegetation. Human activities have been proven to significantly affect the composition and stability of local vegetation, thereby influencing the abundance and species composition of endophytic fungi. Therefore, reducing disturbances to native vegetation and enhancing the diversity of endophytic fungi are key factors in the conservation of the critically endangered plant *A. speciosum.*

### 3.3. Influence of Tissues on Endophytic Fungal Abundance and Species Composition

The diversity, species composition, and relative abundance of endophytic fungal communities vary significantly among different tissues and exhibit tissue specificity [52,53]. The distribution of endophytic fungi within plant tissues is influenced by several factors, such as nutrients, salinity, chemical composition, and environmental conditions of the host plants [54]. In our study, it was found that the diversity of endophytic fungi in the underground parts of *A. speciosum* and *A. aureum* was significantly higher than that in the aboveground parts. Therefore, it is essential to prioritize the conservation of the underground parts of *A. speciosum* and the soil closely associated with the plant’s underground parts. As *A. speciosum* grows under mangrove forests, it is more susceptible to tidal changes than *A. aureum*, which grows at the edge of mangrove forests. Without well-developed mud anchored by the extensive root systems of mangroves, *A. speciosum* would struggle to survive.

### 3.4. Endophytic Fungi and Mangrove Conservation

Fungi require nutrients and energy for their life activities, making their survival environment, such as temperature, humidity, and external disturbances, influential over their life processes. Various human activities inevitably alter the habitat of fungi [55]. The type of plant community primarily determines the composition of fungal communities, and there is a positive correlation between microbial diversity and plant diversity [56].

The tourism industry has made significant contributions to the economic development of China and the world while also exerting a substantial impact on ecological environment and biodiversity [57]. The development of tourism exacerbates anthropogenic activities, which can directly or indirectly affect plants, microorganisms, and other organisms. Endangered plants, mangroves, and nature reserves are important tourism resources that are influenced by human activities [58]. Microorganisms play a crucial role in nutrient cycling, energy conversion, and maintaining ecological balance within ecosystems. They are highly sensitive to environmental changes caused by human activities and are subject to strong disturbances from tourism [59,60].

The tourism industry plays a crucial role in the development of Hainan Province, but its growth also leads to increased anthropogenic activities that pose potential threats to the ecological environment. In terms of the diversity of epiphytic fungi in *A. speciosum* and *A. aureum*, it was observed that the richness of endophytic fungi in heavily human-impacted regions like Wenchang and Haikou was lower compared to that in Zhanjiang. Additionally, the diversity of endophytic fungi in *A. speciosum* was found to be lower than that in *A. aureum*. This lower diversity of endophytic fungi in *A. speciosum* may indicate a weaker capacity to adapt to environmental changes, which could contribute to its endangered status. The impact of tourism on fungal communities in the Dabie Mountains National Forest Park is exacerbated by anthropogenic disturbances, which are increasing in intensity. In areas with low disturbance levels, a total of 896 OTUs were identified. However, under moderate disturbance conditions, this number decreased to 530 OTUs, and in high disturbance areas, only 336 OTUs were observed. As anthropogenic disturbances continue to intensify, there is a gradual decline in fungal abundance [61]. Previous studies have also compared the diversity of endophytic fungal communities in rural and urban forests undergoing urbanization and anthropogenic disturbance. Such disturbances result in forest fragmentation, leading to changes in host plants and a reduction in endophytic fungal diversity [51]. Furthermore, a comparison of fungi in disturbed and undisturbed areas reveals a decrease in fungal community richness in the disturbed regions [62].

To safeguard the microbial environment, mangroves, and endangered plants, it is imperative to establish the rational zoning of areas for human activities and minimize any intentional or unintentional damage caused by human actions. Protective regulations should be implemented to deter destructive behaviors driven by economic interests. Effective communication and education efforts should be undertaken to raise public awareness regarding the importance of protecting microorganisms, endangered plants, and mangroves. Moreover, it is crucial to enhance people’s understanding and awareness of the significant ecosystem services provided by mangrove wetlands and promote the development of sustainable solutions for their management, protection, and utilization.

## 4. Materials and Methods

### 4.1. Study Area and Sample Collection

Samples of *A. speciosum* and *A. aureum* were collected in May 2023 from Zhanjiang, Haikou, and Wenchang in China (Figure 11). Samples from Zhanjiang were collected from Jilongshan, a remote island in Zhanjiang with extensive native mangroves. Samples from Haikou were collected from Dongzhaigang Mangrove Nature Reserve, where *A. speciosum* was introduced from Qinglangang in Wenchang. Samples from Wenchang were collected from Qinglangang Mangrove Reserve. The sampling sites in Haikou and Wenchang were relatively close to residential areas and had a higher number of tourists.

Different tissues of the plants were collected from these three locations, with three replicates for each sample. Samples were cleaned with distilled water, dried with tissue paper to remove surface moisture, and then quickly placed in drikold. The following processes were conducted within 12 h. Samples were cleaned with sterile water, immersed in 75% ethanol for 2–3 min, rinsed again with sterile water, immersed in 1% sodium hypochlorite for 5 min, and finally rinsed with sterile water for 3–5 min. Then samples were placed in centrifuge tubes and stored in a freezer at –80 °C. Once all the samples were collected and processed, they were sent to Guangdong MegGene Technology Co., Ltd. (Guangzhou, China). for ITS sequencing [22,63].

### 4.2. DNA Extraction and PCR Amplification

Samples were weighed to 0.1 g, and the genomic DNA of endophytic fungi was extracted with a DNA extraction kit (Plant DNA Extraction Mini Kit B). The purity and concentration of the endophytic fungi were determined using NanoDrop One. (Thermo Fisher Scientific, Waltham, MA, USA). The qualified samples were subjected to PCR amplification. The ITS1-2 fragment primers (forward primer sequence: CTTGGTCATTTAGAGGAAGTAA, reverse primer sequence: GCTGCGTTCTTCATCGATGC) were used for sequencing. The genomic DNA extracted was set as the template. Based on the selected sequencing region, PCR amplification was performed using specific primers with barcodes and TaKaRa Premix Taq ^®^ Version 2.0 (TaKaRa Biotechnology Co., Dalian, China). The PCR reaction system consisted of 2× Premix Taq (25 μL), Primer-F (10 μM) (1 μL), Primer-R (10 μM) (1 μL), template DNA (50 ng), and nuclease-free water (added to 50 μL). PCR amplification was performed using the BioRad S1000 PCR machine (Bio-Rad Laboratory, Hercules, CA, USA). The PCR reaction conditions were as follows: pre-denaturation at 94 °C for 5 min, denaturation at 94 °C for 30 s, annealing at 52 °C for 30 s, extension at 72 °C for 30 s, a total of 30 cycles, and a final extension at 72 °C for 10 min. The amplified PCR products were purified and stored at 4 °C. Each analysis was repeated three times, and the PCR products of the same sample were mixed. The fragment length and concentration of the PCR products were determined using 1% agarose gel electrophoresis. Samples with main bands in the range of 250–400 bp were used for further experiments. The concentrations of the PCR products were compared using GeneTools Analysis Software (Version 4.03.05.0, SynGene), and the required volume for each sample was calculated based on the principle of equal mass. The PCR products were mixed. The PCR mixture was recovered using the E.Z.N.A.^®^ Gel Extraction Kit (Omega, MA, USA), and the target DNA fragments were eluted with TE buffer. Library construction was performed according to the standard protocol of the NEBNext^®^ Ultra™ II DNA Library Prep Kit for Illumina^®^ (New England Biolabs, Ipswich, MA, USA). The amplified amplicon library was sequenced using the Illumina Nova 6000 platform with PE250 sequencing.

### 4.3. Data Analysis

With fastp 0.14.1 (https://github.com/OpenGene/fastp (accessed on 5 October 2023)), the paired-end raw reads data were subjected to sliding window quality trimming. Simultaneously, the primer sequences at the ends of the sequences were removed using cutadapt software (https://github.com/marcelm/cutadapt/ (accessed on 15 October 2023)) to obtain the quality-controlled paired-end clean reads. For paired-end sequencing data, the overlap relationship between PE reads was utilized to filter out non-matching tags and obtain the original concatenated sequences (raw tags) using usearch -fastq_mergepairs V10 (http://www.drive5.com/usearch/ (accessed on 22 October 2023)).

OTU clustering was performed using UPARSE. The usearch-sintax tool was used to align each representative sequence of the OTUs with the Unite v8.0 database to obtain species annotation information (confidence threshold of 0.8), facilitating the understanding of the species origin of all sequences. The species annotation results were classified into seven hierarchical levels (kingdom, phylum, class, order, family, genus, species). R language was used for statistical analysis of unique and shared species, community species composition, and species abundance clustering analysis. The representative sequences of the top 30 OTUs based on relative abundance were selected, and a phylogenetic tree was constructed using FastTree 2.1.11 software. The confidence of each OTU annotation was annotated based on its relative abundance and the species of its representative sequence, and visualization was performed using ggtree 2.4.1 software. Based on the OTU abundance table, usearch -alpha_div V10 (http://www.drive5.com/usearch/ (accessed on 9 November 2023)) was used to calculate diversity indices and rarefaction curves for species diversity. R v3.5.1 software was used for analysis of inter-group differences in alpha diversity indices using Kruskal–Wallis rank sum tests or one-way ANOVA. The vegan package in R language was used to calculate the beta diversity of species.

Community similarity refers to the degree of similarity between different community structures [64]. There are two methods used to represent it, namely, shared species similarity and species composition similarity [65]. The Jaccard similarity index represents the degree of species composition similarity [66], and its calculation formula is
$$C_{j}=\frac{j}{a+b-j}$$

The symbol $C_{j}$ represents the similarity coefficient of different species in a community. The symbols a and b represent the number of species in two different communities. The symbol j represents the number of identical species in two different communities. When the value of $C_{j}$ is between 0 and 0.25, it indicates a very dissimilar relationship. When the value of $C_{j}$ is between 0.25 and 0.5, it indicates a moderately dissimilar relationship. When the value of $C_{j}$ is between 0.5 and 0.75, it indicates a moderately similar relationship. When the value of $C_{j}$ is between 0.75 and 1, it indicates a highly similar relationship.

## 5. Conclusions

Endophytic fungi are an important component of mangrove ecosystems, forming mutualistic symbiotic relationships with host plants. The endophytic fungi associated with the mangrove ferns *A. speciosum* and *A. aureum* exhibit significant differences in abundance, species composition, geographic distribution, and tissue specificity. The endangered fern *A. speciosum* has lower richness and complexity of endophytic fungi compared to the widely distributed *A. aureum*. The low diversity of endophytic fungi could be one of the main factors contributing to the endangerment of *A. speciosum*. Therefore, reducing human disturbance, protecting local vegetation, and paying attention to the diversity of endophytic fungi in the underground parts of plants can help improve the overall diversity of endophytic fungi in *A. speciosum*, thus enhancing the ability of *A. speciosum* to adapt to environmental changes. Monitoring the dynamic changes of endophytic fungi should be a future research direction for the conservation of endangered mangrove plants and mangrove ecosystems.

## Figures and Tables

**Figure 1 plants-13-00685-f001:**
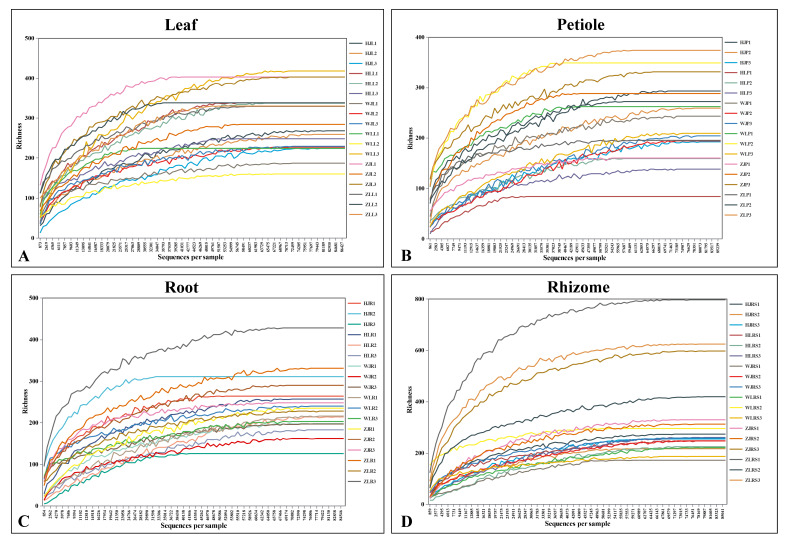
The sparse curves of 72 endophytic fungal samples. (**A**) The sparse curves of 18 samples of endophytic fungi within leaves. (**B**) The sparse curves of 18 samples of endophytic fungi in leaf petioles. (**C**) The sparse curves of 18 root endophytic fungal samples. (**D**) The sparse curves of 18 samples of endophytic fungi in rhizome. H, W, and Z represent Haikou, Wenchang, and Zhanjiang, respectively. J and L represent *A. speciosum* and *A. aureum*, respectively. The third letters L, P, R, and RS represent leaf, petiole, root, and rhizome, respectively.

**Figure 2 plants-13-00685-f002:**
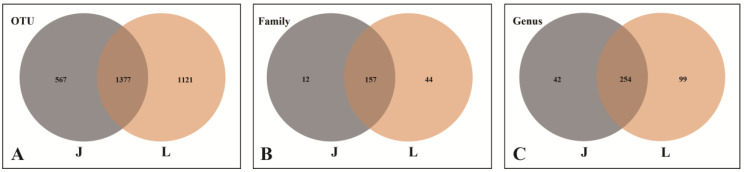
Venn diagrams of the common and unique endophytic fungi. (**A**) At the OTU level. (**B**) At the family level. (**C**) At the genus level. J represents *A. speciosum*, and L represents *A. aureum*.

**Figure 3 plants-13-00685-f003:**
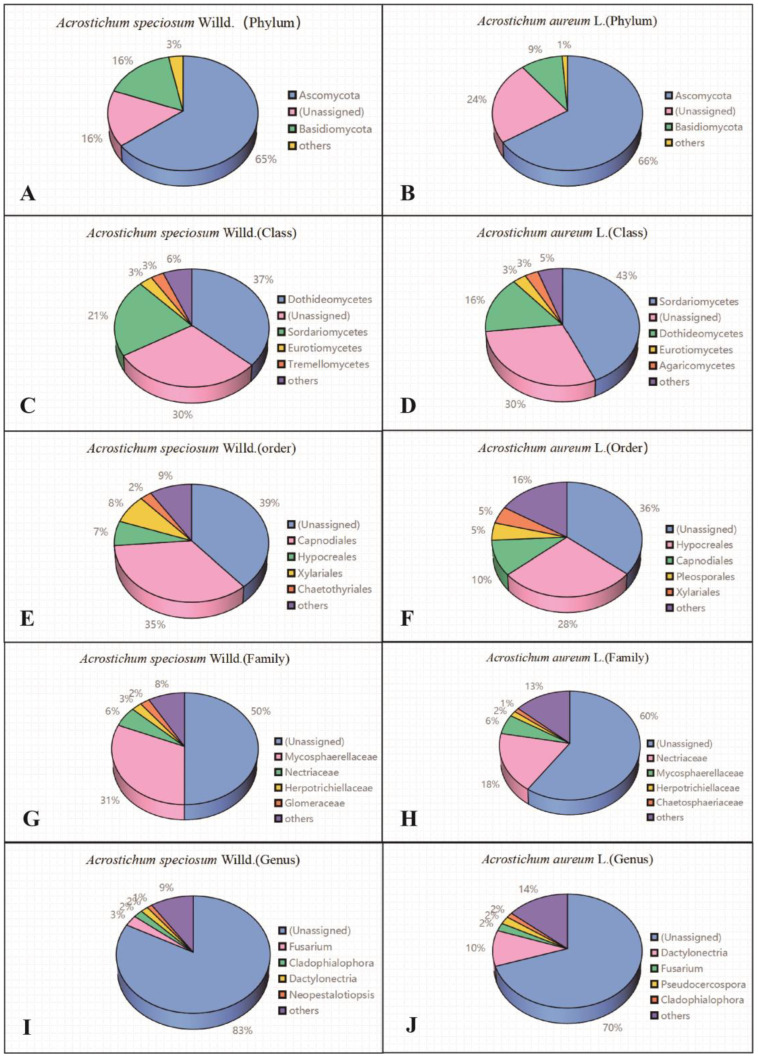
Composition of endophytic fungi in *A. speciosum* and *A*. *aureum (***A**,**B**) at the phylum level; (**C**,**D**) at the class level; (**E**,**F**) at the order level; (**G**,**H**) at the family level; (**I**,**J**) at the genus level.

**Figure 4 plants-13-00685-f004:**
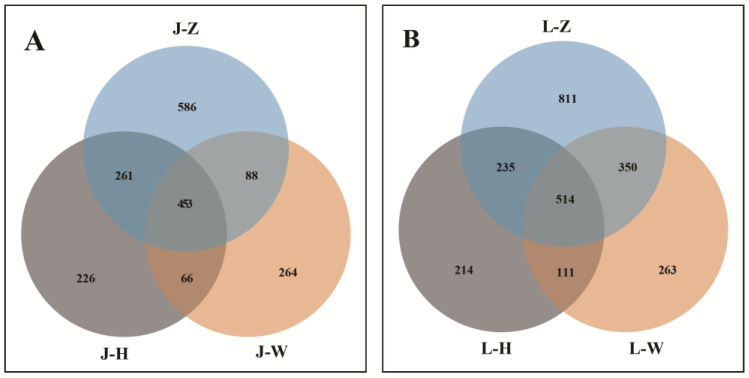
The common and unique endophytic fungi from three locations at toutOTU level ((**A**): *A. speciosum*; (**B**): *A. aureum*). J represents *A. speciosum*, and L represents *A. aureum*; Z represents Zhanjiang, H represents Haikou, and W represents Wenchang.

**Figure 5 plants-13-00685-f005:**
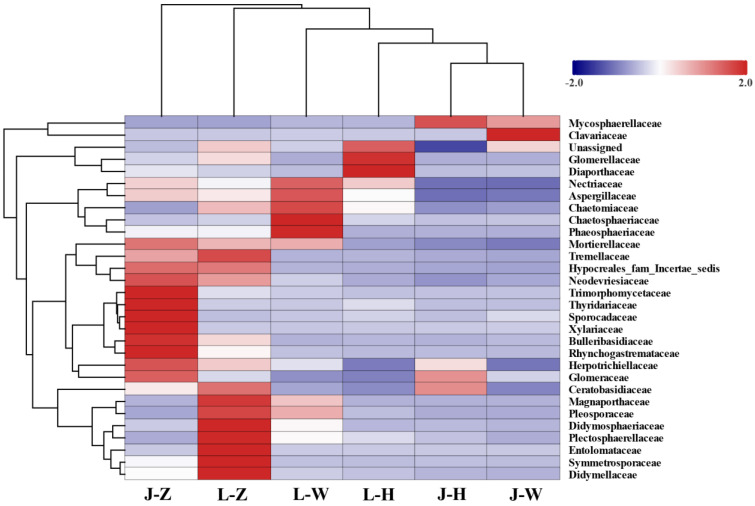
The richness of endophytic fungi of the top 30 families. The horizontal axis represents species and regions, and the vertical axis represents families of endophytic fungi. The clustering tree on the left represents the species clustering tree; the clustering tree on the top represents the species and region clustering tree. J represents *A. speciosum*, and L represents *A. aureum*; Z represents Zhanjiang, H represents Haikou, and W represents Wenchang.

**Figure 6 plants-13-00685-f006:**
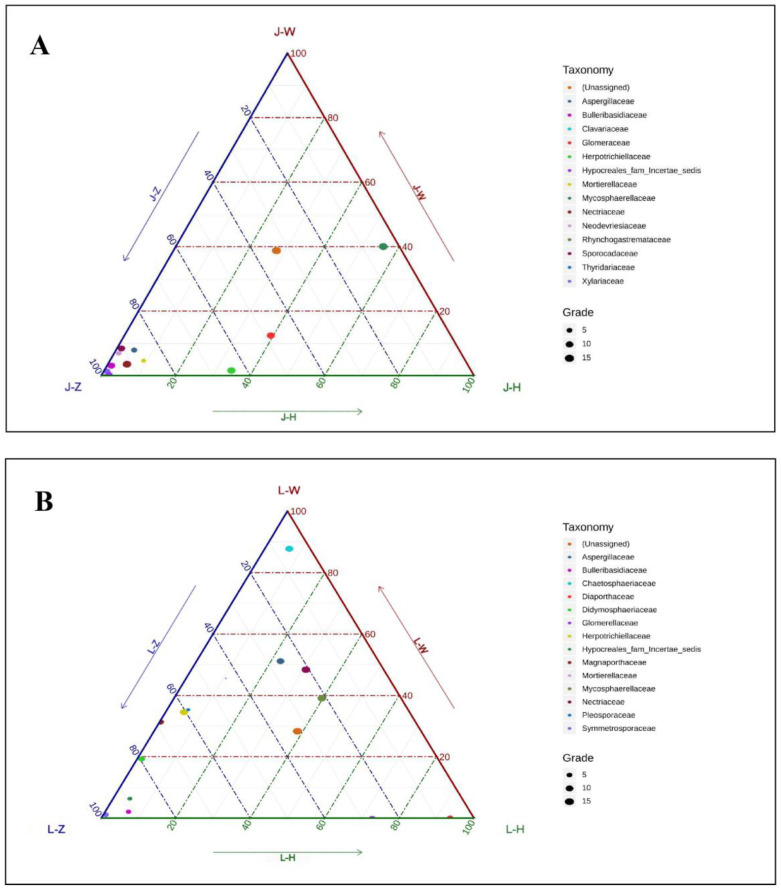
Ternary phase diagram of endophytic fungi of the top 15 families ((**A**): *A. speciosum;* (**B**): *A. aureum*). J represents *A. speciosum*, and L represents *A. aureum*; Z represents Zhanjiang, H represents Haikou, and W represents Wenchang.

**Figure 7 plants-13-00685-f007:**
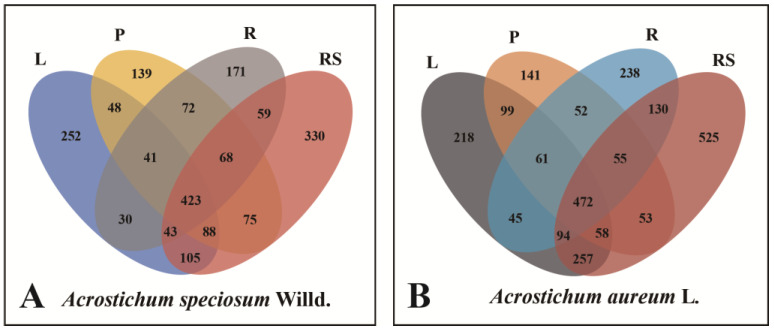
Venn diagrams of endophytic fungi in different tissues at the OTU level ((**A**): *A. speciosum;* (**B**): *A. aureum*). L stands for leaf, P for petiole, R for root, RS for rhizome.

**Figure 8 plants-13-00685-f008:**
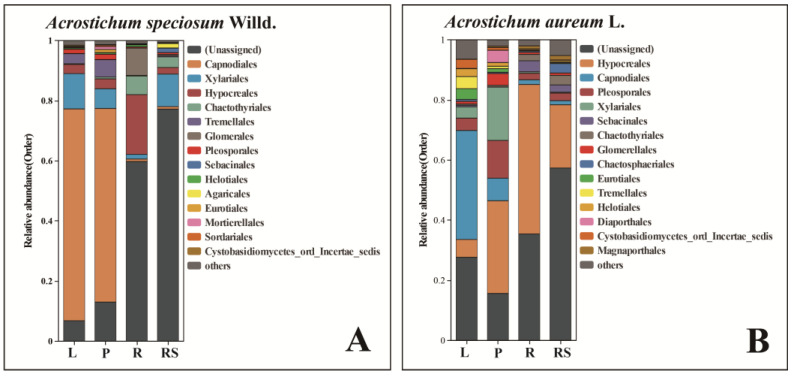
Species composition of endophytic fungi at the order level ((**A**): *A. speciosum;* (**B**): *A. aureum*). L stands for leaf, P for petiole, R for root, and RS for rhizome.

**Figure 9 plants-13-00685-f009:**
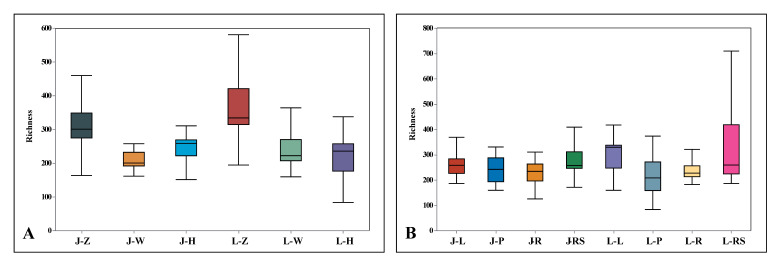
(**A**) The endophytic fungi richness in different locations at the OTU level. (**B**) The endophytic fungi richness in different plant tissues at the OTU level. J stands for *A. speciosum*, and L stands for *A. aureum*; Z stands for Zhanjiang, with W for Wenchang and H for Haikou; L stands for leaf, with P for petiole, R for root, and RS for rhizome.

**Figure 10 plants-13-00685-f010:**
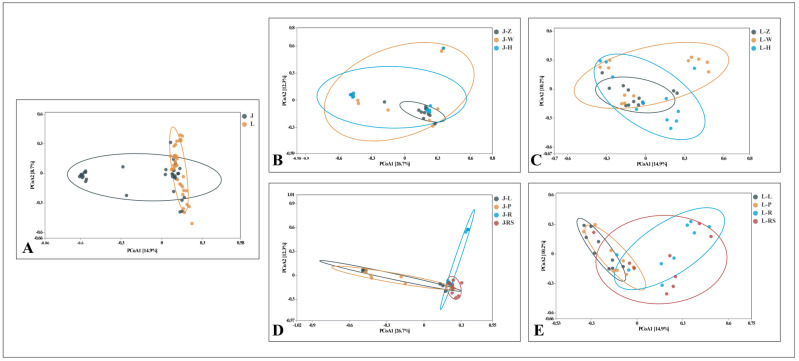
PCoA of endophytic fungi at the OTU level. (**A**) Between two mangrove fern species; (**B**) among three locations of *A. speciosum*; (**C**) among three locations of *A. aureum*; (**D**) among four tissues of *A. speciosum*; (**E**) among four tissues of *A. aureum*. J stands for *A. speciosum*, and L stands for for *A. aureum*; Z stands for Zhanjiang, with W for Wenchang and H for Haikou; L stands for leaf, with P for petiole, R for root, and RS for rhizome.

**Figure 11 plants-13-00685-f011:**
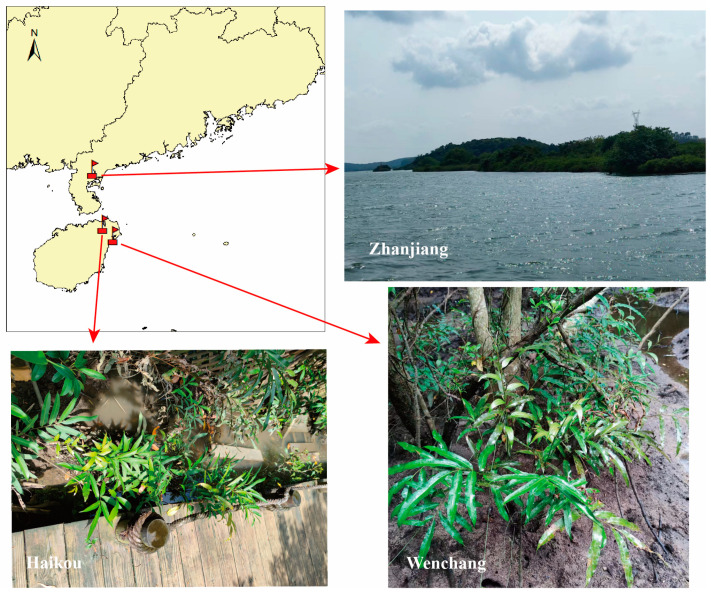
Distribution of sample locations. 
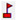
 represents the collecting sites of the samples. The sampling site at Jilong Mountain in Zhanjiang is located at 21°25′45.47″ N, 110°22′30.28″ E. The sampling site at Qinglan Port in Wenchang is located at 19°37′33.07″ N, 110°50′20.89″ E. The sampling site at Dongzhai Port in Haikou is located at 19°56′52.43″ N, 110°34′47.95″ E.

**Table 1 plants-13-00685-t001:** Alpha diversity of endophytic fungi. J stands for *A. speciosum*, and L stands for *A. aureum*; Z stands for Zhanjiang, with W for Wenchang and H for Haikou.

Sample	Simpson	Shannon_e	Chao1	ACE	Pielou
J	0.437	1.789	261.725	335.293	0.321
L	0.270	2.437	289.922	365.543	0.433
J-Z	0.271	2.533	325.175	390.015	0.443
J-W	0.503	1.363	212.783	287.691	0.254
J-H	0.537	1.472	247.217	328.172	0.264
L-Z	0.205	2.913	391.125	463.072	0.496
L-W	0.243	2.528	251.008	321.304	0.458
L-H	0.362	1.870	227.633	312.253	0.344

**Table 2 plants-13-00685-t002:** Community similarity of endophytic fungi from three locations. J stands for *A. speciosum*, and L stands for for *A. aureum*; Z stands for Zhanjiang, with W for Wenchang and H for Haikou.

	J-Z	J-W	J-H	L-Z	L-W
J-W	46.52%				
J-H	53.99%	49.55%			
L-Z	54.86%	39.64%	46.77%		
L-W	49.49%	49.00%	44.66%	55.89%	
L-H	58.82%	51.46%	54.85%	49.11%	51.49%

**Table 3 plants-13-00685-t003:** Similarities of endophytic fungi in four tissues. J stands for *A. speciosum*, and L stands for *A. aureum*; L stands for leaf, with P for petiole, R for root, and RS for rhizome.

	J-L	J-P	J-R	J-RS	L-L	L-P	L-R
J-P	57.98%						
J-R	56.83%	60.54%					
J-RS	58.54%	52.53%	53.09%				
L-L	52.19%	58.71%	50.57%	49.31%			
L-P	56.60%	60.17%	62.79%	51.18%	58.08%		
L-R	53.18%	52.79%	54.55%	53.99%	55.33%	53.23%	
L-RS	45.78%	47.39%	46.10%	49.04%	57.69%	46.69%	55.46%

## Data Availability

Data are contained within the article and Appendix A.

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
