# Peer review of "Endophytic Fungal Diversity of Mangrove Ferns Acrostichum speciosum and A. aureum in China"

_plants, 2024, doi:10.3390/plants13050685_

Round 1

Reviewer 1 Report

Comments and Suggestions for Authors

Major comments

1. The introduction is dense with information. Consider breaking it down into more focused paragraphs, each addressing a specific aspect (e.g., general importance of mangroves, decline in mangrove areas, role of microbial communities, focus on endophytic fungi, and the specifics regarding A. speciosum and A. aureum).

2. Authors mentions the decline in mangrove areas and the importance of studying endophytic fungi in the introduction. However, What are the key questions regarding the endophytic fungi in A. speciosum and A. aureum that this study aims to address?

3. The introduction could benefit from focus on how understanding endophytic fungal diversity can contribute to the conservation of A. speciosum and mangrove ecosystems

4. Authors effectively establish mangroves as rich sources of endophytic fungi, highlighting their global significance. However, the connection between the abundance of these fungi and the specific characteristics of mangroves could be explored further.

5. The potential reasons for the endangered status of A. speciosum linked to its lower fungal diversity are interesting. However, this section could be strengthened by discussing how enhancing fungal diversity could contribute to conservation strategies.

6. The impact of human activities on fungal communities is a valid point. but how this could be preserved, add some information by suggesting specific conservation measures to mitigate these effects.

Minor comments

Improve the unclear figures (1, 3, 5, 6, and 10) for better visual information.

Specify the extraction kit used for transparency and reproducibility.

Clarify the criteria for determining sample suitability for PCR.

Give more details on what "main bands within the normal range" means for sample selection.

Mention the specific version of the Unite database used in the study.

Change "Discuss" to "Discussion" for consistency and clarity in section headings.

Taxonomic names should be italicized

Comments on the Quality of English Language

Moderate editing of English language required

Author Response

Comments and Suggestions for Authors

Major comments

  1. The introduction is dense with information. Consider breaking it down into more focused paragraphs, each addressing a specific aspect (e.g., general importance of mangroves, decline in mangrove areas, role of microbial communities, focus on endophytic fungi, and the specifics regarding speciosumand A. aureum).

Thanks for the reviewer's suggestion, it has been segmented according to your suggestion, modified and added a small part of content.

  1. Authors mentions the decline in mangrove areas and the importance of studying endophytic fungi in the introduction. However, What are the key questions regarding the endophytic fungi in speciosumand A. aureum that this study aims to address?

Thank you very much for the reviewer's suggestion. This study aims to compare the differences between the endophytic fungi of A. speciosum and A. aureum. From the perspective of endophytic fungi, this study aims to analyze why both A. speciosum and A. aureum belong to Acrostichum, and why A. speciosum are extremely dangerous plants in China, while A. aureum are widely analyzed.This paper puts forward some reasonable suggestions for the protection of A. speciosum and provides a new idea for the protection of mangroves.

  1. The introduction could benefit from focus on how understanding endophytic fungal diversity can contribute to the conservation of speciosumand mangrove ecosystems
  2. Authors effectively establish mangroves as rich sources of endophytic fungi, highlighting their global significance. However, the connection between the abundance of these fungi and the specific characteristics of mangroves could be explored further.

Thanks very much for the reviewer's valuable suggestions. In combination with the third and fourth questions, the relationship between microorganisms, mangroves and endophytic fungi has been introduced in detail in the introduction. In Discussion 3.4, it is also mentioned that microbial diversity is positively correlated with mangrove plant diversity.

  1. The potential reasons for the endangered status of speciosumlinked to its lower fungal diversity are interesting. However, this section could be strengthened by discussing how enhancing fungal diversity could contribute to conservation strategies.
  2. The impact of human activities on fungal communities is a valid point. but how this could be preserved, add some information by suggesting specific conservation measures to mitigate these effects.

Thanks for the reviewer's suggestions. In combination with the fifth and sixth questions, I added a subsection (3.4) to the discussion, which expounded the impact of human activities on endophytic fungi and mangroves and put forward some feasible suggestions.

Minor comments

  1. Improve the unclear figures (1, 3, 5, 6, and 10) for better visual information.

Thanks to the reviewer, the clarity of the picture in the article has been adjusted, but the clarity of the picture inserted in word seems to be reduced, and a clear original picture with modification has been provided to the editor.

  1. Specify the extraction kit used for transparency and reproducibility.

Thanks for the reviewer's suggestion, the Plant DNA Extraction Mini Kit B kit was used in this experiment, which has been added in the article in L558

  1. Clarify the criteria for determining sample suitability for PCR.

Thanks for the reviewer's suggestion, generally speaking, primers have their own PCR product length, and it is feasible to amplify clear bands within the length range of the pair of primers.

  1. Give more details on what "main bands within the normal range" means for sample selection.

Thanks to the reviewer's suggestion, the main band in the normal range is between 250bp-400bp, and relevant content has been added in the article, in line 575.

  1. Mention the specific version of the Unite database used in the study.

Thanks to the reviewer's suggestion, the v8.0 version used in this study has been added in 595.

  1. Change "Discuss" to "Discussion" for consistency and clarity in section headings.

Thanks to the reviewer's suggestion, the revision has been made in the article, in line 422.

  1. Taxonomic names should be italicized

Thank you for the valuable suggestions from the reviewer. We have carefully reviewed the entire article and made revisions accordingly. The Latin names of species or genera of endophytic fungi and plants have been italicized, while the Latin names of endophytic fungi at other taxonomic levels have not been italicized. My supervisor mentioned that Latin names other than species and genera do not need to be italicized. We would appreciate it if the reviewer could provide feedback on whether our viewpoint is correct. If our viewpoint is incorrect, we will make immediate revisions. We sincerely appreciate the reviewer's input.

Reviewer 2 Report

Comments and Suggestions for Authors

This article highlighted the importance of Endophytic fungal diversity of mangrove ferns Acrostichum L. in China. To improve the quality of a manuscript, there are some comments:

L3 Acrostichum speciosum and A. aureum or comparison between shade preferred and light preferred fern

L19 Provide full name of OTU

L21 A bit vague. Please provide numeric values or other expressions. 

L41 Provide references regarding this.

L43 Also, it could be better to provide references with some unit of areas (i.e. ha) or percents.

L44 How about mainland China and Taiwan? any recent reports related to mangroves diminishing?

L44 remove dot.

L46 I know this paper focuses on mangroves in China, but it might be better to mention a bit which countries are showing high productivity or outstanding ecosystem services on a global scale like this: China and all over the world such as Indonesia, New Zealand or etc..

L49 in salinity and nutrient levels

L50 biogeochemical or soil chemical properties?

L54 To solid your words, please provide references as well.

L66 ITS means Internal transcribed spacer? can you please provide full name of it for readers?

L73 Reference

L76 Any references regarding Acrostichum's growth, carbon sequestration, survival rate or other ecosystem services in China or other regions?

L101 full name initially

L115 Please adjust all figures size or resolution in this manuscript.

L390 Discussion 

L484 Can you provide latitude and longitude in the captions?

L562 I think this part is too broad. I think it might be better to explain how much you did quantify endophytic fungal diversity by region and explain low diversity of endophytic fungi could result in a weak capacity of A. speciosum toadapt to environmental changes, thus leading to its endangered status.

Author Response

(1)L3 Acrostichum speciosum and A. aureum or comparison between shade preferred and light preferred fern

Many thanks to the reviewer, it has been modified according to your suggestions in the article L2-L3

(2)L19 Provide full name of OTU

Thanks to the reviewer's suggestion, the full name of OTU is operational taxonomic unit, which has been modified in L19

(3)L21 A bit vague. Please provide numeric values or other expressions.

Thanks to the reviewer's suggestion, the revision has been made in L20-L21 of the article

  • L41 Provide references regarding this.

(5)L43 Also, it could be better to provide references with some unit of areas (i.e. ha) or percents.

(6)L44 How about mainland China and Taiwan? any recent reports related to mangroves diminishing?

Thanks for the reviewer's suggestion. In combination with (4), (5) and (6) Q, this part has been modified in combination with the latest reports and literatures, and relevant literatures have been cited. Please review the content in lines 45-60 of the article.

(7)L44 remove dot.

Thanks to the reviewer, I will carefully check the whole article, and there will be no similar low-level mistakes.

(8)L46 I know this paper focuses on mangroves in China, but it might be better to mention a bit which countries are showing high productivity or outstanding ecosystem services on a global scale like this: China and all over the world such as Indonesia, New Zealand or etc..

Thanks for the reviewer's suggestion, it is China and all over the world, which has been revised in the article, lines 61-62 of the article.

(9)L49 in salinity and nutrient levels

Thanks for the reviewer's suggestion, it has been revised in the article, in lines 65-66.

(10)L50 biogeochemical or soil chemical properties?

Thanks for the reviewer's suggestion, it has been revised in the article in L67

(11)L54 To solid your words, please provide references as well.

Thanks to the reviewer's suggestion, relevant references have been added

(12)L66 ITS means Internal transcribed spacer? can you please provide full name of it for readers?

Thanks to the reviewer's suggestion, ITS means Internal transcribed spacer, has been modified in the article, in L84.

(13)L73 Reference

We greatly appreciate the valuable feedback from the reviewer. The available literature on A. speciosum is scarce both domestically and internationally. Previous reports have indicated that A. speciosum is classified under the genus Acrostichum, family Acrostichaceae. However, it is important to note that these reports are dated back to five years ago. According to the current information obtained from the Plant Wisdom database, A. speciosum is classified under the genus Acrostichum, family Pteridaceae. Thus, for the purpose of this paper, we will adopt the classification provided by the Plant Wisdom database as the authoritative reference. Accordingly, we have included the relevant link (https://www.iplant.cn/info/Acrostichum speciosum). Additionally, we have included the appropriate references pertaining to A. aureum in the manuscript, specifically in Section L93.

(14)L76 Any references regarding Acrostichum's growth, carbon sequestration, survival rate or other ecosystem services in China or other regions?

Thank you very much for the reviewer's suggestion, and relevant references have been cited in the article.

(15)L101 full name initially

Thanks to the reviewer, changes have been made in the article, in L122

(16)L115 Please adjust all figures size or resolution in this manuscript.

Thanks to the reviewer, the clarity of the picture in the article has been adjusted, but the clarity of the picture inserted in word seems to be reduced, and a clear original picture with modification has been provided to the editor.

(17)L390 Discussion

Thanks to the reviewer, changes have been made in the article, in L422

(18)L484 Can you provide latitude and longitude in the captions?

Thanks to the reviewer, the relevant content has been added to the article according to your suggestion, in L550-554

  • L562 I think this part is too broad. I think it might be better to explain how much you did quantify endophytic fungal diversity by region and explain low diversity of endophytic fungi could result in a weak capacity of speciosumtoadapt to environmental changes, thus leading to its endangered status.

Thanks for the reviewer's suggestion. Combined with your and another reviewer's suggestion, I added 3.4 in the discussion part of the article. I am not sure whether this can answer the reviewer's question, please review it.

Round 2

Reviewer 1 Report

Comments and Suggestions for Authors

Satisfied with revised draft

Comments on the Quality of English Language

Minor editing of English language required

Author Response

1.Minor editing of English language required

Thank you very much for reviewing the manuscript, which has been polished according to your suggestions.

Reviewer 2 Report

Comments and Suggestions for Authors

I think the authors tried to reflect most of my comments to improve the quality of the manuscripts. However, it is the comments for the discussion part.

L487 How can you connect and/or was it related to your results and findings? Were there any correlation data between diversity of endophytic fungi in plants and anthropogenic impact in Hainan Province based on your data or references (numeric values)?

Author Response

1.L487 How can you connect and/or was it related to your results and findings? Were there any correlation data between diversity of endophytic fungi in plants and anthropogenic impact in Hainan Province based on your data or references (numeric values)?

Thank you very much for the review, which has been revised according to your suggestions in lines 520-531 of the article.

Round 3

Reviewer 2 Report

Comments and Suggestions for Authors

I believe the authors tried to improve the quality of manuscript, but here are some comments:

L14 I think I already mentioned you do use the full name before mentioning the abbreviation. Please check all manuscripts carefully.

L518 What are the physical characteristics of different plant species? Can you explain it in the manuscript?

L573 Human activities

Author Response

  1. L14 I think I already mentioned you do use the full name before mentioning the abbreviation. Please check all manuscripts carefully.

I am very grateful for the suggestions from the reviewer. I have carefully examined the entire article and made the necessary revisions.

  1. L518 What are the physical characteristics of different plant species? Can you explain it in the manuscript?

I am very grateful for the reviewer's suggestion. In this paragraph, the main purpose is to describe the influence of host plant and plant nutrient composition on the abundance of endophytic fungi. The previous description may not be accurate, and relevant parts have been modified. I would like to express my sincere appreciation to the reviewer for their critical comments.

  1. L573 Human activities

Thank you to the reviewer for their valuable feedback. The suggested revisions have been incorporated into the manuscript, specifically at line 500.
